# Advancements and Perspectives in Biodegradable Polyester Elastomers: Toward Sustainable and High-Performance Materials

**DOI:** 10.3390/ijms26020727

**Published:** 2025-01-16

**Authors:** Lisheng Tang, Xiaoyan He, Ran Huang

**Affiliations:** 1Academy for Engineering and Technology, Yiwu Research Institute, Zhuhai Fudan Innovation Institute, Fudan University, Shanghai 200433, China; tangls@rizt.ac.cn (L.T.); xiaoyanhe@rizt.ac.cn (X.H.); 2Center for Innovation and Entrepreneurship, Taizhou Institute of Zhejiang University, Taizhou 318000, China

**Keywords:** biodegradable, elastomer, polyester, glycerol, citric acid, open-bond cross-linkage

## Abstract

While the traditional rubber industry faces the severe pressure of environmental pollution and carbon emissions, bio-based and biodegradable elastomers have become a hot topic in the field and drawn intensive research interest. Inspired by polyester resin, incorporating polyol or polycarboxylic acid as a branching unit into aliphatic polyester and/or introducing a monomer with a C=C bond to provide open-bond cross-linking in the fashion of common vulcanization to form three-dimensional network structures are two mainstream strategies for designing biodegradable polyester elastomers (BPEs). Both methods encounter more or fewer problems, such as poor mechanical and thermal properties due to the easy hydrolysis of the ester bond and space hinderance, or the potential harm of the remaining degraded small molecules with olefin bonds. This article provides an overview of recent endeavors aimed at addressing these challenges and prospects the probable future advancements in the field.

## 1. Introduction

Elastomer (rubber) is a type of polymer material with a low modulus and high elasticity that can quickly restore its initial shape after removing external stress. Due to the high elasticity, resilience, toughness, and ductility of elastomers, they have irreplaceable application value in almost all fields of global economic life and industry [1,2,3,4,5,6]. However, so far, almost all commercial synthetic elastomers are non-degradable, and a large portion of them are synthesized from fossil resources. Like the “white pollution” caused by plastics, the elastomer industry also faces severe environmental pollution and carbon emissions, and there are almost no microorganisms in nature that can degrade traditional rubber. On the other hand, biomass is a type of important renewable resource, and bio-based polymers have lower greenhouse gas emissions compared to polymers from fossil sources [7,8,9]. Therefore, attention on biodegradable and renewable resources has become more and more intensive, and developing biodegradable elastomers to reduce carbon emissions and dependence on fossil resources has become a hot topic in the field. Aliphatic polyester with a good biodegradability has become a research hotspot as a key type of degradable material [10]. Linear polyesters such as polylactide (PLA) [11,12], polyglycolide (PGA) [13,14], and polycaprolactone (PCL) [15,16] have been developing rapidly as typical biodegradable materials. Inspired by these linear polyesters, polyester elastomers with three-dimensional network structures have also been continuously reported [17,18,19]. This material possesses the capacity to rapidly restore its deformation under conditions with substantial stress–strain due to its cross-linked three-dimensional network structure. Additionally, the presence of ester bonds in aliphatic polyester elastomers contributes to their biodegradability. Nevertheless, the mechanical properties and thermal stability of polyester are compromised by the ester linkage’s susceptibility to hydrolysis and spatial hindrance. This issue is of particular concern in the rubber industry. Consequently, there is an ongoing endeavor to develop novel biodegradable polyester elastomers (BPEs) with an enhanced performance and reinforce them to ensure their practical utility. In this manuscript, we are going to undergo a brief review on these efforts in recent years, aiming to harness the natural environmentally friendly and low-carbon advantages of such materials.

## 2. Bio-Based Polyester Elastomers

Bio-based diols and dicarboxylic acids, such as itaconic acid, 1,4-butanediol, succinic acid, sebacic acid, and other “Top 12” bio-based monomers [20], can all be obtained through the fermentation of biomass resources. The design and synthesis of biodegradable polyester elastomers (BPEs) primarily follow two distinct strategies, as illustrated in Figure [1]. In situ cross-linking polyester elastomers are synthesized using the first approach (Figure 1a), which involves the use of multifunctional monomers, such as polyols or polycarboxylic acids, as branching units for in situ cross-linking. Through esterification polycondensation, these monomers form dendritic prepolymers containing a high density of functional groups. These prepolymers are subsequently cross-linked to create polyesters with a three-dimensional (3D) network structure. This method is particularly advantageous and is commonly used in biomedical applications due to its excellent biocompatibility and non-biological toxicity [21,22,23].

Polyolefin cross-linked polyester elastomers are synthesized using the second approach (Figure 1b), which involves the open-bond cross-linking of unsaturated polyesters. In this strategy, cross-linking is achieved through the open-bond reactions of unsaturated C=C bonds, akin to the classical process of rubber vulcanization, resulting in the formation of a robust 3D network. In this pathway, linear unsaturated polyesters are first synthesized through esterification polycondensation, incorporating unsaturated alkyl groups (C=C bonds) into the molecular chains. These unsaturated bonds act as reactive sites for cross-linking, which is achieved through chemical reactions, such as free radical polymerization, to form a 3D network. This strategy enables the design of BPEs with tunable mechanical properties and an enhanced elasticity by adjusting the molecular composition and cross-linking density, to create materials with an excellent elasticity and mechanical robustness [24].

Both methods aim to address the dual challenges of biodegradability and mechanical performance, offering versatile solutions for applications in biomedical devices, environmental materials, and beyond. These approaches highlight the importance of functional monomers and cross-linking strategies in achieving the desired structural and functional characteristics of BPEs.

### 2.1. In Situ Cross-Linked Polyester Elastomer

In situ cross-linked polyester elastomers are polymerized from multifunctional monomers as the branching unit, and glycerol and citric acid are the two most widely studied monomers.

#### 2.1.1. Glycerol Based Polyester Elastomer

Glycerol is the simplest renewable polyol among natural resources, commonly known as 1,2,3-propaniol. It is a fundamental component of many natural lipids and can be obtained from various sources such as animals, plants, and microorganisms. It has been approved by the US Food and Drug Administration (FDA) for medical use [25]. Glycerol-based polyesters are synthesized from glycerol and various aliphatic dicarboxylic/polycarboxylic acids of different chain lengths, all of which can be derivatives of renewable resources, to form a three-dimensional network structure [26,27,28,29]. Among all polyesters synthesized from glycerol, poly(glycerol sebacate) (PGS) is, so far, the most prominent polyester used in biomedical applications [17,30,31], and is the first generation of thermosetting polyester elastomers [32].

PGS is usually obtained by the further structural modification, curing, blending, and molding of its prepolymer (pre-PGS). Different synthesis methods can lead to various structures and properties. According to present research reports, the main methods for synthesizing pre-PGS include vacuum polycondensation, enzyme-catalyzed synthesis, and microwave-assisted synthesis [22]. Polycondensation with a reduced pressure is the earliest reported and most promising method for synthesizing pre-PGS, which involves esterification at high temperatures under an inert gas atmosphere [32,33,34,35]. The enzymatic synthesis of PGS has also been intensively studied due to its mild conditions, high catalytic efficiency, and selectivity [36,37]. Perin and Felisberti [38] used immobilized Candida Antarctica Lipase B (CALB) to produce PGS under mild reaction conditions, and studied its kinetics, chain growth, and branching behavior under different conditions (solvent, temperature, CALB amount, and reagent feed ratio). Ning et al. [39] showed that, compared to self-catalyzed condensation polymerization, immobilized Antarctic Candida Lipase B (N435) catalyzed the synthesis of PGS with a higher molecular weight. Microwave radiation is also an effective way to shorten the pre-polymerization time of PGS [40]. Lau et al. [41] successfully prepared a biomaterial composed of PGS and β-tricalcium phosphate nanoparticles. Tevlek et al. [42] obtained PGS with a good elasticity (212.75 ± 37.25% elongation and 0.09 ± 0.03 MPa Young’s modulus) by microwave pre-polymerization and then curing in a vacuum oven (150 °C, 12 h). Compared with traditional melt polycondensation, without changing the molar ratio of glycerol to sebacic acid, microwave synthesis produced highly branched pre-PGS, which helped to form crosslinked PGS in a shorter time [43].

The most commonly used method for curing pre-PGS into PGS with a three-dimensional structure is thermal crosslinking. It can react for a shorter time at higher temperatures (which, however, may lead to PGS degradation) or for a longer time (even a few days) at a lower temperature (120–150 °C). The crosslinking temperature and time greatly affect the degree of crosslinking, thereby affecting the mechanical properties of the polymer [44,45]. Chemical crosslinking is also one of the ways to achieve pre-PGS curing. Golbaten Mofrad et al. [46] used Sn(Oct)_2_ as a catalyst to continuously stir pre-PGS and different amounts of HDI in a DMSO–DMF (7:3) mixed solvent at 55 °C for 15 min to synthesize PGS-U with different crosslinking densities. The crosslinking conditions were milder and the crosslinking speed was faster. Introducing citric acid as a crosslinking agent in pre-PGS has also been proven to be a feasible method for promoting crosslinking. By adding a small amount of citric acid crosslinking agent, PGS-derived elastomers with similar properties to PGS can be obtained with only a small fraction of the original curing time [47]. Another way is to functionalize pre-PGS to make it UV curable. The commonly used method by researchers is to introduce components containing double bonds (acrylation [48,49], norbornenylation [50], etc.) on the PGS chain. Under the action of a photoinitiator and appropriate wavelengths of light, functionalized pre-PGS can quickly crosslink at room temperature and pressure to form elastomers with three-dimensional network structures. Table 1 summarizes key findings, advantages, and applications, showcasing the evolution of strategies to optimize PGS’s properties for biomedical and engineering applications, highlighting advancements in polycondensation, enzymatic synthesis, microwave-assisted techniques, and chemical or UV cross-linking.

Poly(glycerol succinate) (PGSU) is a novel polyester material that can be synthesized through the polycondensation of glycerol and succinic acid, a dicarboxylic acid. Agach et al. [51] synthesized PGSU using bio-based succinic acid and glycerol as raw materials through a solvent-free and catalyst-free method. The experimental results indicated that PGSU exhibited non-ecotoxicity and a good biodegradability. Nakiou et al. [52] synthesized PGSU and evaluated its application as a coating for bioglass scaffolds. They explored the preparation, characterization, and potential of PGSU in biomedical applications, offering new insights and ideas for research on biodegradable materials. Shukla et al. [53] synthesized PGSU with varying chain lengths by incorporating different diacids. They investigated its thermal stability and hydrolytic degradation, providing both experimental data and theoretical support for the optimization of performance and the expansion of applications for biodegradable polyester materials. Mahtabi et al. [54] combined PGSU with PLA and n-HA to prepare a novel nanocomposite scaffold based on PLA/PGSU/n-HA through salt leaching. Comprehensive data were presented on the scaffold’s characteristics and cell responses, with PLA70PGSU30H5 demonstrating a good performance. This research provided valuable insights into the application of PGSU in tissue engineering. Godinho et al. [55] synthesized a series of poly(glycerol co-diacid) polymers through condensation reactions using glycerol, sebacic acid, and succinic acid in various molar ratios, including poly(glycerol sebacate) (PGS), poly(glycerol succinate) (PGSu), and poly(glycerol sebacate co-succinate) (PGSSu) with seven different ratios. The study systematically examined the influence of the succinic acid ratio on the polymer properties, providing a foundation for designing poly(glycerol co-diacid) polymers with specific characteristics and guiding future optimization for various biomedical applications.

In addition to sebacic acid and succinic acid, a multitude of other diacids can serve as monomers in the synthesis of poly(glycerol co-diacids). These include aliphatic diacids with varying chain lengths ranging from C4 to C14 [26,27], 2,5-furancarboxylic acids [56], as well as other unsaturated fatty diacids [57].

#### 2.1.2. Citric-Acid-Based Polyester Elastomer

Similar to glycerol, citric acid (CA) is also a standard bio-based multifunctional monomer that can be used to form three-dimensional polymers. The thermosetting elastomer obtained by the melt polycondensation reaction of citric acid and glycol monomers exhibits good biodegradability and biocompatibility. The α- and β-carboxyl of diol and CA forms a three-dimensional network at 160 °C under nitrogen protection [18]. Common examples of binary alcohols include 1,6-hexanediol, 1,8-octanediol, and 1,10-decanediol, among which the extensively investigated variant is the poly(citrate-1,8-octanediol) ester (POC). In 2004, Yang et al. [58] obtained a POC bioelastomer through the direct melt polycondensation of citric acid and 1,8-octanediol. It had an ester bond structure similar to linear polyester, as well as a good elasticity, biocompatibility, biodegradability, and flexibility. Citrate-based polyester elastomers (CBEs) are generally used in the biomedical field. Due to the different reactivity of α- and β-Carboxyl groups, the crosslinking degree, mechanical properties, and degradability of CBE can be adjusted by changing the molar ratio and reaction kinetics between CA and diols [58,59,60,61].

The potential for enhancing the performance of CBE through the exclusive manipulation of polymerization process parameters is constrained. Consequently, in recent years, scholars have undertaken several endeavors to enhance the modulus and tensile strength of CBEs [62,63]. Guo et al. [64] synthesized functional POC prepolymers using azides and alkyne glycols, and then obtained POC elastomers with a tensile strength higher than 20 MPa by thermal synchronous double cross-linking the POC prepolymers. The composite film of POC/chitosan (CS) produced by Zeimaran et al. [65] demonstrated an extensive range of mechanical properties, which were dependent on the chitosan concentration. The ester groups present in POC have the ability to undergo a reaction with the amino and hydroxyl groups in CS at a temperature of 80 °C over a period of 7 days. This reaction leads to the formation of a network structure, which facilitates the dispersion of CS within the POC matrix. As a result, the mechanical characteristics of POC/CS composites are enhanced. A film, which incorporateed 30% CS, exhibited a tensile strength of 5.87 MPa and a modulus of 15.96 MPa. The tensile strength and elastic modulus exhibited a positive correlation with the rise in the CS concentration, however, the elongation at break demonstrated a negative correlation. Ren et al. [66] developed a series of POC-based composite materials using bioactive glass (BG) and phytic-acid-modified bioactive glass (PSC). The results indicated that the calcium in BG can enhance the crosslinking of POC/BG composites by forming dicarboxylic acid calcium bridges, thereby improving their mechanical properties. Compared with POC/BG composites, POC/PSC composites exhibited significantly better mechanical properties. By adding 70 wt% PSC, the compressive strength of POC/PSC composites can be increased to about 50 MPa and the modulus can reach 1.3 ± 0.1 GPa. Li et al. [67] developed multifunctional poly(citrate-siloxane) hybrid elastomers (PCS-SN) reinforced by silicon dioxide nanoparticles (SNs) through in situ nanoparticle formation technology. SNs with a uniform size and spherical morphology were uniformly distributed in the PCS polymer matrix, and the addition of SNs significantly enhanced the elastic properties of PCS. In addition to the aforementioned bio-based glycerol and CA, xylitol [68,69], sorbitol [70,71], and mannitol [72] are also widely studied bio-based multifunctional monomers. Table 2 outlines various synthesis methods for and innovations in citric-acid-based polyester elastomers (CBEs), highlighting synthesis methods, key findings, and applications. The focus spans advancements in mechanical properties, biodegradability, biocompatibility, and specific applications in tissue engineering, bone regeneration, and environmentally friendly materials. The versatility of CBEs positions them as promising materials for biomedical and sustainable applications.

### 2.2. Polyolefin Cross-Linked Polyester Elastomers

In situ cross-linked polyester elastomers synthesized from multifunctional monomers are typically soft, biocompatible, and biodegradable and are mainly used as biomedical materials, especially in soft tissue engineering. Although the mechanical properties of these in situ crosslinked polyester elastomers can generally meet the application needs of biomedical materials, the demand for traditional engineering materials is far from sufficient.

In order to develop BPEs with better mechanical properties, open-bond polyolefin crosslinked polyester elastomers have been developed along the same trend, which have a processing and cross-linking process similar to traditional rubber. Open-bond cross-linkage polyester elastomers introduce components containing an unsaturated bond C=C into the polyester molecular chain, which provides crosslinking sites for the formation of three-dimensional network structures. The design strategy for synthesizing cross-linkable polyester mainly involves the following two aspects: (1) breaking the regularity of polyester molecular chains through multi-component copolymerization and/or introducing side groups, thereby disrupting crystallization and obtaining amorphous polyester; (2) introducing monomers containing C=C during the copolymerization and condensation process to provide reaction sites for open-bond cross-linking. Multicomponent copolymerization can reduce chemical and geometric regularity, reduce the glass transition temperature, and inhibit crystallization. The first synthesized bio-based elastomer is poly-(1,4-butanediol/1,3-propanediol/sebacic acid/itaconic acid/succinate)-ester formed by the condensation of five monomers, with an average molecular weight (M_n_) ranging from 32,952 to 52,529 g/mol. The long chain of sebacic acid increases the mobility of the molecular chain and reduces the density of ester groups in polyester. Itaconic acid containing olefin side groups makes open-bond crosslinking possible. Amorphous polyester elastomers can be obtained through vulcanization and crosslinking with diisopropylene peroxide (DCP) [24]. Afterwards, more novel bio-based polyester elastomers were developed, with potential application value in many fields [73,74]. Wang et al. [74] synthesized aliphatic unsaturated polyester (M_n_ = 2300 g/mol) from dicarboxylic acids (such as succinic acid, sebacic acid, and itaconic acid) and diols (such as 1,3-propanediol and 1,4-butanediol) derived from biomass resources through melt polycondensation, and then obtained elastic polyester nanoparticles by emulsification and radiation crosslinking. 2,3-butanediol is a commonly used biologically branched diol, in which the presence of two methyl side groups can effectively disrupt the crystallization of polyester chains. Hu et al. [75] synthesized another type of polyester (PBPSSI, M_n_ = 15,000–33,000 g/mol) from 1,3-propanediol, 2,3-butanediol, succinic acid, sebacic acid, and itaconic acid. The obtained bio-based elastomer exhibited an excellent thermal stability and biocompatibility, and the tensile strength of PBPSSI can be greatly improved with the reinforcement of silica. Kang et al. [76] synthesized a bio-based polyester elastomer with a combination of itaconic acid, succinic acid, 1,3-propanediol, 1,4-butanediol, and sebacic acid. Then, the PLA was mixed with a suitable substance for reinforcement. At an elastic component of 11.5 vol.%, the notch impact strength of the composite material exhibited a notable enhancement, rising from 2.4 to 10.3 kJ/m^2^, reflecting a substantial increase of 330%. Hu et al. [77] reported a novel bio-based polyester (PLBSI) synthesized by the condensation polymerization of lactic acid, sebacic acid, itaconic acid, and 1,4-butanediol. Introducing lactic acid into PLBSI provided it with a lactic acid structure similar to PLA, and it had a good compatibility with PLA. With an increase in the PLBSI content, the stress–strain behavior of PLBSI/PLATPV shifted from plastic to elastic characteristics. In addition to toughening PLA, bio-based polyester elastomers also have broad development prospects in the fields of conductivity and shape memory. Tang et al. [78] synthesized a bio-based polyester elastomer through the condensation reaction of bio-based diol and diacid and grafted it onto graphene to obtain a polyester/graphene composite material. When the loading amount of graphene was 0.33 vol.%, the conductivity of polyester/graphene composite material could reach 1.06 S/m. Kang et al. [73] synthesized a poly(butanediol/isosorbide/sebacate/itaconate) copolyester. The copolyesters obtained by optimizing the content of isosorbitol exhibited an excellent shape memory performance, shape fixation, and nearly a 100% shape recovery rate.

One challenge in obtaining BPEs is that the cross-linkage does not guarantee elasticity, and a specific monomer combination design is necessary to tune the expected elasticity of the copolyester. Kang et al. [73] developed a bio-based shape memory polymer for biomedical applications, utilizing 1,4-butanediol, isosorbide, sebacic acid, and itaconic acid to create a range of cross-linked copolyesters with outstanding shape recovery properties. However, these PBISI copolyesters did not exhibit elastomeric behavior at room temperature, likely due to the absence of longer-chain monomers that provide a greater flexibility, which is essential for elasticity. Recently, Tang et al. [79] developed a similar polyester PBHIIS with the introduction of 1,6-hexanediol to reduce the regularity of the copolyester, resulting a constant elastomer under room temperature with a decent tensile strength and elongation rate.

Additionally, efforts seeking the combination of in situ cross-linkage and polyolefin cross-linkage have also been progressed to take the advantage of a dual network. Tang et al. [80] reported a biodegradable poly(ethylene glycol-glycerol-itaconate-sebacate) (PEGIS) copolyester elastomer, which employed glycerol as a multifunctional branching unit for in situ cross-linkage and itaconic acid for C=C open bonding linkage.

To summarize, open-bond cross-linkage polyester elastomers not only broaden the application range of bio-based polyester, but also provide new ideas for the development of new bio-based polyester composite materials. The key features of the above studies are summarized in Table 3.

## 3. Conclusions and Perspective

The present article provides a progressive review of the various synthesis techniques, structural considerations, mechanical and thermal properties, and prospective applications pertaining to BPEs. The advancement of biodegradable elastomers has the potential to mitigate the reliance on fossil resources, minimize waste generation, and decrease carbon emissions.

Novel structures and properties of crosslinked BPEs can be achieved by the utilization of bio-based ethanol and carboxylic acids. In situ cross-linkages can be formed via multifunctional monomers, such as CA and glycerol. Bio-based polyester elastomers usually exhibit weak mechanical properties and are commonly used in biomedical applications. Chemical modification and blending with fillers are expected to significantly improve the performance of bio-based polyester elastomers and expand their application range. Polyester elastomers that are open-bond crosslinked utilize unsaturated double bonds as sites for vulcanization. These elastomers have varying properties and suitability for diverse applications, owing to the wide range of bio-based ethanol and carboxylic acids available.

Nanofillers such as carbon black and silica can effectively improve mechanical properties. Due to their good compatibility with polar plastics such as PLA, open-bond cross-linkage polyester elastomers have a significant effect on toughening polar plastics and can be employed in the preparation of thermoplastic vulcanizates (TPVs). In addition, they can also be applied to fields such as conductive elastomers and shape memory materials. Customizing specific properties via the design of the molecular structure is the future direction of bio-based open-bond cross-linkage polyester elastomers, as well as high-performance and multifunctional modification through formula optimization.

The wide variety of bio-based monomers provides the possibility to develop bio-based elastomers with various characteristics. However, low production, high costs, and a weaker performance than traditional rubber are common issues faced by bio-based elastomers. To achieve the sustainable and green production of elastomers, it is necessary to explore low-cost technologies for the large-scale production of bio-based monomers, develop bio-based elastomers for high value-added applications, and accelerate their commercialization. There is a pressing need for more advanced practical techniques to enhance their thermal and mechanical properties. While they may not completely replace regular rubber, having a sufficient strength is crucial in many practical situations.

Furthermore, the degradation rate of BPEs is generally fast, which limits their application in certain fields. This degradation rate needs to be adjusted according to different application scenarios, as well as considering sustainability and economy. If the service life of a BPE is too short, it not only limits the application, but also increases the usage cost. Further research and technological innovation are necessary to achieve a designable and controllable degradation rate.

On the other hand, one of the challenges associated with the open-bond cross-linking strategy is the inevitable introduction of non-degradable small molecule fragments during the cross-linking process, particularly when polyolefin-based monomers are used. These fragments can accumulate in the environment, potentially undermining the core goal of biodegradability. While this contradiction may seem to limit the sustainability of this approach, it highlights an important aspect of material design: the balance between achieving advanced material properties and minimizing environmental impact. Researchers can address this challenge by optimizing the composition and cross-linking agents to reduce the formation of non-degradable byproducts. For instance, integrating bio-based monomers with degradable linkages or designing cross-linking systems that break down into benign, environmentally friendly compounds can mitigate this issue. Additionally, applications where material recovery and controlled degradation are feasible, such as biomedical devices, provide a controlled pathway for responsible disposal or recycling. Ultimately, this trade-off encourages innovation in creating materials that fulfill demanding performance criteria while aligning with sustainable principles, reflecting the delicate balance between technological advancement and ecological stewardship.

The potential applications of BPEs are extensive; nevertheless, their full realization is hindered by certain problems that need to be addressed. We anticipate that forthcoming technical improvements will offer a greater array of ecologically friendly and sustainable options.

## Figures and Tables

**Figure 1 ijms-26-00727-f001:**
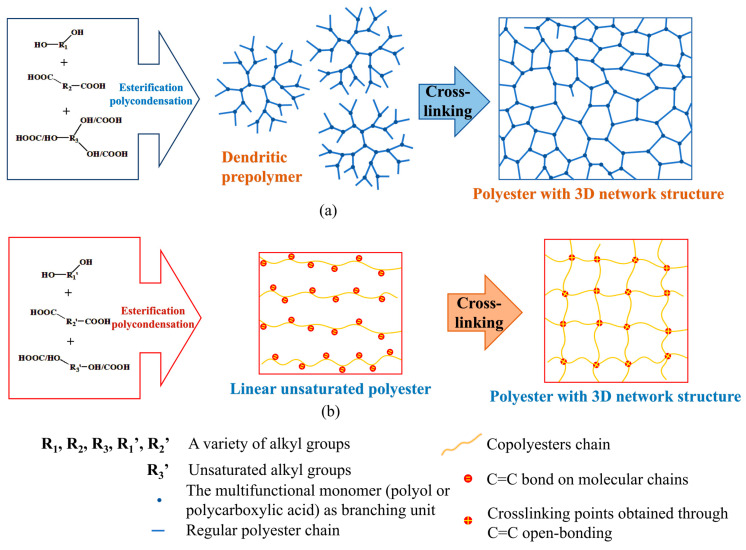
The scheme to achieve biodegradable polyester elastomers: (**a**) in situ cross-linking polyester elastomers—this method involves the esterification polycondensation of multifunctional monomers such as polyols or polycarboxylic acids to form dendritic prepolymers. These prepolymers are subsequently cross-linked to create polyesters with a dense 3D network structure, offering enhanced elasticity and mechanical properties. (**b**) Open-bond cross-linkage polyester elastomers—this approach starts with the synthesis of linear unsaturated polyesters via esterification polycondensation. The incorporation of unsaturated alkyl groups (C=C bonds) into the molecular chains allows for chemical cross-linking through C=C open bonding, resulting in polyesters with tailored 3D network structures and mechanical performance.

**Table 1 ijms-26-00727-t001:** Overview of synthesis and cross-linking methods for poly(glycerol sebacate) (PGS).

Study	Synthesis Method	Key Findings	Applications/Remarks
Wang et al. [32]; Xuan et al. [33]; Wu et al. [34]; Piszko et al. [35]	Polycondensation under reduced pressure	Early and promising method for synthesizing pre-PGS at high temperatures in an inert gas atmosphere.	Provides a strong foundation for producing PGS.
Godinho et al. [36]; Lang et al. [37]	Enzymatic synthesis	Mild conditions, high catalytic efficiency, and selectivity.	Suitable for environmentally friendly synthesis of PGS prepolymers.
Perin and Felisberti [38]	Enzymatic synthesis with CALB	Explored reaction kinetics, chain growth, and branching under various conditions.	Enhanced understanding of enzymatic polymerization parameters.
Ning et al. [39]	Enzymatic synthesis with N435	Achieved higher-molecular-weight PGS compared to self-catalyzed polymerization.	Improved mechanical properties of PGS prepolymer.
Aydin et al. [40]	Microwave-assisted synthesis	Reduced pre-polymerization time for PGS.	Efficient and energy-saving synthesis technique.
Lau et al. [41]	Microwave synthesis of PGS/β-TCP composites	Successfully incorporated β-tricalcium phosphate nanoparticles into PGS.	Biomedical applications such as bone tissue engineering.
Tevlek et al. [42]	Microwave pre-polymerization and curing	Achieved PGS with good elasticity (elongation: 212.75 ± 37.25%, Young’s modulus: 0.09 ± 0.03 MPa).	Demonstrated the effectiveness of microwave synthesis for elastomer preparation.
Lau et al. [43]	Microwave-assisted synthesis	Produced highly branched pre-PGS, enabling faster cross-linking.	Enhances the production efficiency of crosslinked PGS.
Conejero-García et al. [44]	Thermal crosslinking	Correlated curing parameters with physicochemical properties of PGS.	Provides insights into controlling PGS mechanical properties through thermal treatment.
Golbaten-Mofrad et al. [46]	Chemical cross-linking with Sn(Oct)2	Synthesized PGS-U with varied crosslinking densities using a mild and fast method.	Allows fine-tuning of PGS properties for specific applications.
Risley et al. [47]	Citric acid cross-linking	Introduced citric acid to accelerate cross-linking, reducing curing time.	Achieved similar properties to PGS in a fraction of the original time.
Pashneh-Tala et al. [48]; Becerril-Rodriguez et al. [49]	UV-curable pre-PGS	Functionalized PGS with double bonds (acrylation) for UV-curable crosslinking.	Quick and efficient elastomer formation under mild conditions.
Yeh et al. [50]	UV-curable pre-PGS	Functionalized PGS with double bonds (norbornenylation) for UV-curable crosslinking.	Quick and efficient elastomer formation under mild conditions.

**Table 2 ijms-26-00727-t002:** Summary of advancements in citric-acid-based polyester elastomers (CBEs).

Study	Synthesis Method	Key Findings	Applications/Remarks
Chon et al. [18]	Melt polycondensation	Citric acid and glycol monomers form a 3D network at 160 °C under nitrogen protection; good biodegradability and biocompatibility.	Versatile elastomer suitable for biomedical applications.
Yang et al. [58]; Wang et al. [59]; Yu et al. [60]; Koper et al. [61]	Direct melt polycondensation	Developed poly(citrate-1,8-octanediol) ester (POC) with excellent elasticity, biocompatibility, and flexibility.	Widely used in tissue engineering and biomedical applications.
Zhu et al. [62]	Electrospinning	Fabricated fibrous mats from poly(1,8-octanediol citrate) for tissue engineering.	Soft tissue scaffolds with biodegradable properties.
Liang et al. [63]	Biodegradable nanocomposite fabrication	Created nanocomposites using chitin nanocrystals and poly(caprolactone-diol citrate) elastomers.	Applications in biomedical engineering and biodegradable materials.
Guo et al. [64]	Thermal synchronous double cross-linking	Achieved POC elastomers with tensile strength exceeding 20 MPa by using azides and alkyne glycols.	Enhanced mechanical performance for biomedical and engineering applications.
Zeimaran et al. [65]	Composite synthesis with chitosan (CS)	Created POC/CS composites with adjustable mechanical properties; tensile strength reached 5.87 MPa.	Blend films for tissue engineering, mechanical properties dependent on CS concentration.
Ren et al. [66]	POC-based composite with bioactive glass	Enhanced crosslinking with calcium in bioactive glass, achieving a compressive strength of ~50 MPa.	Applications in bone regeneration and implants.
Li et al. [67]	In-situ nanoparticle reinforcement	Developed poly(citrate-siloxane) hybrid elastomers reinforced with silicon dioxide nanoparticles.	Multifunctional properties for bioimaging and bone tissue regeneration.
Piątek-Hnat et al. [68]; Firoozi et al. [69]	Melt polycondensation with xylitol	Synthesized biodegradable xylitol–sebacate copolyesters with tunable properties.	Potential for biomedical applications.
Geeti et al. [70]; Piątek-Hnat et al. [71]	Melt polycondensation with Sorbitol	Developed environmentally benign bio-based waterborne polyesters with good thermal and biodegradation properties.	Applications in coatings and environmentally friendly materials.
Rahmani et al. [72]	Development of poly(mannitol sebacate) nanofibers	Fabricated poly(mannitol sebacate)/poly(lactic acid) nanofibrous scaffolds for tissue engineering.	Suitable for soft tissue applications with biodegradable characteristics.

**Table 3 ijms-26-00727-t003:** Summary of recent advancements in the development of Polyolefin cross-linked polyester elastomers.

Study	Polyester Type	Synthesis Method	Key Monomers	Properties/Applications
Wei et al. [24]	PBPSIS Amorphous Polyester Elastomer	Condensation, vulcanization, and cross-linking with diisopropylene peroxide (DCP)	1,4-butanediol, 1,3-propanediol, sebacic acid, itaconic acid, succinate	Initial development of cross-linked elastomers through classical vulcanization methods. Average molecular weight (M_n_) ranging from 32,952 to 52,529 g/mol.
Kang et al. [73]	PBISI (Bio-based Shape Memory Polyester)	Copolymerization, and cross-linking	1,4-butanediol, iso-sorbide, sebacic ac-id, itaconic acid	Outstanding shape recovery proper-ties, not elastomeric at room temper-ature due to lack of long-chain monomers.
Wang et al. [74]	Aliphatic Unsaturated Polyester	Melt polycondensation, emulsification, and radiation cross-linking	Succinic acid, sebacic acid, itaconic acid, 1,3-propanediol, 1,4-butanediol	Weight loss ratio of 52.3% within 5 days in the presence of lipase), low glass transition temperature of about −55 °C.
Hu et al. [75]	PBPSSI (Bio-based Elastomer)	Melt polycondensation and silica reinforcement	1,3-propanediol, 2,3-butanediol, succinic acid, sebacic acid, itaconic acid	Excellent thermal stability, biocompatibility, improved tensile strength through silica reinforcement.
Kang et al. [76]	Bio-based Polyester Elastomer	Melt polycondensation and PLA reinforcement	Itaconic acid, succinic acid, 1,3-propanediol, 1,4-butanediol, sebacic acid	Enhanced notch impact strength (330% increase), potential for composite reinforcement.
Hu et al. [77]	PLBSI (Polyester Similar to PLA)	Condensation	Lactic acid, sebacic acid, itaconic acid, 1,4-butanediol	Compatible with PLA, stress-strain behavior shifted from plastic to elastic, potential in conductivity and shape memory.
Tang et al. [78]	Polyester/Graphene Composite	Condensation reaction with graphene grafting	Bio-based diols and diacids	Conductivity of 1.06 S/m at 0.33 vol.% graphene, applications in conductive materials.
Tang et al. [79]	PBHIIS (Bio-based Elastomer)	Copolymerization	1,6-hexanediol, isosorbide, sebacic acid, itaconic acid	Constant elastomer behavior at room temperature with good tensile strength and elongation rate.
Tang et al. [80]	PEGIS-BC (Biodegradable Elastomer)	Combination of in-situ and polyolefin cross-linking	Glycerol, itaconic acid, sebacic acid	Dual cross-linking network, strengthened with bacterial cellulose.

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
