# Peer review of "Advancements and Perspectives in Biodegradable Polyester Elastomers: Toward Sustainable and High-Performance Materials"

_ijms, 2025, doi:10.3390/ijms26020727_

Round 1

Reviewer 1 Report

Comments and Suggestions for Authors

The review is well written, but there are some very serious errors. For example, it is unacceptable for a review submitted to a journal like the International Journal of Molecular Sciences with such a high impact factor and citescore to have only 3 topics of discussion, basically highlighting Polyolefin cross-linked polyester elastomers and In-situ cross-linked polyester elastomer, nothing about Poly(ester-urethanes), Poly(ester-ethers), Segmented polyesters is mentioned separately and adequately. The review is very superficial and contributes nothing to scientific progress in the field.

Author Response

3. Point-by-point response to Comments and Suggestions for Authors

Comments 1:  The review is well written, but there are some very serious errors. For example, it is unacceptable for a review submitted to a journal like the International Journal of Molecular Sciences with such a high impact factor and citescore to have only 3 topics of discussion, basically highlighting Polyolefin cross-linked polyester elastomers and In-situ cross-linked polyester elastomer, nothing about Poly(ester-urethanes), Poly(ester-ethers), Segmented polyesters is mentioned separately and adequately. The review is very superficial and contributes nothing to scientific progress in the field.

Response 1:  Thank you for your valuable feedback on my manuscript. We appreciate your insights and the time you took to review my work. We would like to clarify that this submission is intended as a focused review, which is why the content is more concise and the discussions may appear less in-depth. Moreover, about Poly(ester-urethanes) and Poly(ester-ethers), there has been just published an excellent review on another MDPI journal: < Burelo et al., Recent Developments in Synthesis, Properties, Applications and Recycling of Bio-Based Elastomers, Molecules 2024, 29, 387. https://doi.org/10.3390/molecules29020387 >, in which these two types of polymers were intensively discussed, however the polyester elastomers our group focused on was not mentioned too much. Therefore, we decided to write a review for a good compensation, instead of a duplicated work. We sincerely appreciate your suggestions for expanding the scope of the review. Thank you once again for your constructive criticism, and we will take it into consideration for future work.

4. Response to Comments on the Quality of English Language

Point 1: The English could be improved to more clearly express the research.

Response 1:    Thank you for your valuable feedback regarding the clarity of the English in my manuscript. I have carefully reviewed the text and made revisions to enhance clarity and readability. The revision was marked in red in the revised manuscript.

5. Additional clarifications

Reviewer 2 Report

Comments and Suggestions for Authors

The submitted manuscript addresses a very important topic related to biodegradable elastomers. While in the case of thermoplastics the topic is recognized, in elastomers it is a significant problem. The manuscript fits into current research trends. It is a review in nature.

1. The title of the manuscript is consistent with its content.

2. The introduction is well organized. A bit modest but provides a good background for the stated goal.

3. The literature review contains over 60 items. It is well done. It also includes older items, which indicates interest in the topic and its relevance. Of course, the literature could be supplemented with a few more items, but it reflects the background of the work well.

4. The tables and the lists made in them are a good addition. They supplement the significant information contained in the main text.

5. The general approach to the topic of biodegradable elastomers gives the reader the possibility of their own interpretation of the presented theses and solutions. This is shown in a fairly extensive summary and final conclusions.

6. Despite the small size of the manuscript, the authors have actually reviewed the selected element related to biodegradable elastomers. The manuscript can be the beginning of a longer discussion on these materials.

Author Response

3. Point-by-point response to Comments and Suggestions for Authors

Comments 1:  The title of the manuscript is consistent with its content.

Response 1:  I am pleased to hear that you find the title of the manuscript consistent with its content. I aimed to ensure that the title accurately reflects the focus of the review.

Comments 2:  The introduction is well organized. A bit modest but provides a good background for the stated goal.

Response 2:  Thank you for your positive feedback on the organization of the introduction.

Comments 3:  The literature review contains over 60 items. It is well done. It also includes older items, which indicates interest in the topic and its relevance. Of course, the literature could be supplemented with a few more items, but it reflects the background of the work well.

Response 3:  I am glad that you found the literature review comprehensive and reflective of the topic's relevance. I agree with your suggestion to supplement it with a few more recent items to further enhance the background of the work. These have been revised and the revision was marked in red in the revised manuscript.

Comments 4:  The tables and the lists made in them are a good addition. They supplement the significant information contained in the main text.

Response 4:   I appreciate your recognition of the value added by the tables and lists. I aimed to present significant information clearly, and I am glad that you found them helpful.

Comments 5:  The general approach to the topic of biodegradable elastomers gives the reader the possibility of their own interpretation of the presented theses and solutions. This is shown in a fairly extensive summary and final conclusions.

Response 5:  Thank you for your positive comments.

Comments 6:  Despite the small size of the manuscript, the authors have actually reviewed the selected element related to biodegradable elastomers. The manuscript can be the beginning of a longer discussion on these materials.

Response 6:  I appreciate your acknowledgment of the manuscript's contribution to the discussion on biodegradable elastomers. I hope it serves as a foundation for further exploration of these materials.

4. Response to Comments on the Quality of English Language

Point 1: The English could be improved to more clearly express the research.

Response 1:    Thank you for your valuable feedback regarding the clarity of the English in my manuscript. I have carefully reviewed the text and made revisions to enhance clarity and readability. The revision was marked in red in the revised manuscript.

5. Additional clarifications

Reviewer 3 Report

Comments and Suggestions for Authors

I believe that before publishing this manuscript the authors should consider the following issues:

General:

1) Nomenclature - according to IUPC guidelines, polymers with a multiple name should contain brackets in their name. For example, poly(glycerol sebacate), not polyglycerol sebacate.

2) The authors use glycerol/glycerin inconsistently and interchangeably. They should stick to one option for clarity.

3) In Chapter 1 the authors write about 'excellent degradability'. Either you need to explain exactly what this 'excellence' looks like, or you should not use such statements. In some applications this degradability is not appropriate.

4) At the beginning of Chapter 2, the authors write about the 'Top 12'. Please explain exactly where this term comes from and what it includes.

5) The authors write: ‘This method is particularly advantageous for are commonly used in biomedical applications due to their excellent biocompatibility and non-biological toxicity’ – there is lack of something here.

6) In line 105 the authors used some symbol instead of a dot at the end of the sentence.

7) When describing the types of PGS cross-linking, the authors should refer more often to the literature. This is a review paper, so it should be maximally authenticated. For example, when writing about the most common cross-linking method - thermal, the authors refer to only one article. This matter should be refined. There are many publications on these topics (e.g. regarding thermal cross-linking 10.37190/ABB-02208-2023-04)

8) Why did the authors decide to describe only citric-based, sebacic-based, and crosslinked polyolefin elastomers? This should have been clearly explained at the beginning of Chapter 2.

9) Chapter 2.1.1 refers to glycerol-based materials, and 2.1.2 to citric-based materials. There is a big inconsistency, because in one case the authors take one part (from sebacic acid and glycerol), and in the other - the other part (from citric acid and glycol). Moreover, it is an abuse to call chapter 2.1.1. from glycerol, since it concerns only PGS.

Author Response

3. Point-by-point response to Comments and Suggestions for Authors

Comments 1:  Nomenclature - according to IUPC guidelines, polymers with a multiple name should contain brackets in their name. For example, poly(glycerol sebacate), not polyglycerol sebacate.

Response 1:  Thank you for pointing this out. We agree with this comment. Therefore, we have revised the nomenclature throughout the manuscript to follow IUPAC guidelines, ensuring that polymers with multiple names are presented correctly in brackets (e.g., poly(glycerol sebacate)). And the revision was marked in red in the revised manuscript.

Comments 2:  The authors use glycerol/glycerin inconsistently and interchangeably. They should stick to one option for clarity.

Response 2:  Thank you for pointing this out. We acknowledge the inconsistency in using glycerol/glycerin. We have standardized the terminology to consistently use "glycerol" throughout the manuscript for clarity. And the revision was marked in red in the revised manuscript.

Comments 3:  In Chapter 1 the authors write about 'excellent degradability'. Either you need to explain exactly what this 'excellence' looks like, or you should not use such statements. In some applications this degradability is not appropriate.

Response 3:  Thank you for your suggestion. We have revised 'excellent degradability' to 'biodegradability' and marked in red in the revised manuscript.

Comments 4:  At the beginning of Chapter 2, the authors write about the 'Top 12'. Please explain exactly where this term comes from and what it includes.

Response 4:  Thank you for your time. This term comes from the U.S. Department of Energy. Report: Top Value Added Chemicals from Biomass, Volume I: Results of Screening for Potential Candidates from Sugars and Synthesis Gas. 2004. It's available electronically at http://www.osti.gov/bridge. The 'Top 12' includes:

1) Four Carbon 1,4-Diacids (Succinic, Fumaric, and Malic)

2) 2,5-Furan dicarboxylic acid (FDCA)

3) 3-Hydroxy propionic acid (3-HPA)

4) Aspartic acid

5) Glucaric acid

6) Glutamic acid

7) Itaconic acid

8) Levulinic acid

9) 3-Hydroxybutyrolactone

10) Glycerol

11) Sorbitol (Alcohol Sugar of Glucose)

12) Xylitol/arabinitol (Sugar alcohols from xylose and arabinose)

Comments 5:  The authors write: ‘This method is particularly advantageous for are commonly used in biomedical applications due to their excellent biocompatibility and non-biological toxicity’ – there is lack of something here.

Response 5:  Thank you for your comment. We have revised ‘This method is particularly advantageous for are commonly used in biomedical applications due to their excellent biocompatibility and non-biological toxicity’ to ‘This method is particularly advantageous for applications in the biomedical field due to its excellent biocompatibility and minimal biological toxicity’. And the revision was marked in red in the revised manuscript.

Comments 6:  In line 105 the authors used some symbol instead of a dot at the end of the sentence.

Response 6: Thank you for pointing this out. The manuscript has been revised and the revision was marked in red in the revised manuscript.

Comments 7:  When describing the types of PGS cross-linking, the authors should refer more often to the literature. This is a review paper, so it should be maximally authenticated. For example, when writing about the most common cross-linking method - thermal, the authors refer to only one article. This matter should be refined. There are many publications on these topics (e.g. regarding thermal cross-linking 10.37190/ABB-02208-2023-04).

Response 7:  Thank you for your constructive feedback. The publication (10.37190/ABB-02208-2023-04) has been cited in the revised manuscript.

Comments 8:  Why did the authors decide to describe only citric-based, sebacic-based, and crosslinked polyolefin elastomers? This should have been clearly explained at the beginning of Chapter 2.

Response 8:  Thank you for your suggestion. In-situ cross-linking polyester elastomers and polyolefin cross-linked polyester elastomers are biodegradable polyester elastomers synthesized through two different strategies, respectively. And glycerol and citric acid are the two most widely studied monomers to synthesize in-situ cross-linking polyester elastomers. These have been explained and marked in red at the beginning of Chapter 2 and Chapter 2.1.

Comments 9:  Chapter 2.1.1 refers to glycerol-based materials, and 2.1.2 to citric-based materials. There is a big inconsistency, because in one case the authors take one part (from sebacic acid and glycerol), and in the other - the other part (from citric acid and glycol). Moreover, it is an abuse to call chapter 2.1.1. from glycerol, since it concerns only PGS.

Response 9:  Thank you for your constructive feedback. Glycerol based polyesters are synthesized from glycerol and various aliphatic dicarboxylic/polycarboxylic acids of different chain lengths to form three-dimensional network structure. Among all polyesters synthesized from glycerol, PGS is the so far the most prominent polyester used in biomedical applications. So most of the research is focused on PGS. In order to enrich this section, we have added content on other glycerol-based polyesters in the revised manuscript.

4. Response to Comments on the Quality of English Language

Point 1: The English could be improved to more clearly express the research.

Response 1:    Thank you for your valuable feedback regarding the clarity of the English in my manuscript. I have carefully reviewed the text and made revisions to enhance clarity and readability. The revision was marked in red in the revised manuscript.

5. Additional clarifications

Round 2

Reviewer 3 Report

Comments and Suggestions for Authors

Good work on the answers. I have no more comments.